# High-Dose Acetaminophen with *N*-acetylcysteine Rescue Inhibits M2 Polarization of Tumor-Associated Macrophages

**DOI:** 10.3390/cancers15194770

**Published:** 2023-09-28

**Authors:** Allyn Bryan, Pavani Pingali, Martha Joslyn, Howard Li, Tytus Bernas, Jennifer Koblinski, Joseph Landry, Won Sok Lee, Bhaumik Patel, Alexander Neuwelt

**Affiliations:** 1Department of Veterans Affairs, Richmond, VA 23249, USA; 2Department of Veterans Affairs, Charleston, SC 29405, USA; 3Department of Anatomy and Neurobiology, Virginia Commonwealth University, Richmond, VA 23284, USA; 4Department of Pathology, Virginia Commonwealth University, Richmond, VA 23284, USA; 5Department of Human and Molecular Genetics, Virginia Commonwealth University, Richmond, VA 23284, USA; 6Department of Hematology and Oncology, Virginia Commonwealth University, Richmond, VA 23284, USA

**Keywords:** acetaminophen, breast cancer, tumor-associated macrophages

## Abstract

**Simple Summary:**

High dose acetaminophen (AAP) with *N*-acetylcysteine (NAC) rescue has demonstrated promising results in pre-clinical and clinical studies. However, the mechanism is not clearly understood. In the present manuscript, we evaluate the effects of high dose AAP/NAC on the tumor immune microenvironment. We demonstrate that high dose AAP inhibits M2 polarization of tumor associated macrophages at the RNA and protein levels. The anti-tumor activity of high dose AAP is lost in macrophage depleted mice, underlining the physiologic relevance of the findings. While at traditional doses AAP is not thought to have anti-inflammatory activity, our study is the first to describe AAP-induced changes on anti-tumor immunity.

**Abstract:**

High-dose acetaminophen (AAP) with *N*-acetylcysteine (NAC) rescue is among the few treatments that has shown activity in phase I trials without achieving dose-limiting toxicity that has not progressed to evaluation in later line studies. While the anti-tumor effects of AAP/NAC appear not to be mediated by glutathione depletion and free radical injury, the mechanism of anti-tumor effects of AAP/NAC has not been definitively characterized. In vitro, the effects of AAP/NAC were evaluated on bone marrow derived macrophages. Effects of AAP on IL-4/STAT6 (M2) or IFN/LPS/STAT1 (M1) signaling and downstream gene and protein expression were studied. NAC reversed the AAP toxicity in the normal liver but did not reverse AAP cytotoxicity against tumor cells in vitro. AAP/NAC selectively inhibited IL-4-induced STAT6 phosphorylation but not IFN/LPS-induced STAT1 phosphorylation. Downstream, AAP/NAC inhibited IL-4 induction of M2-associated genes and proteins but did not inhibit the IFN/LPS induction of M1-associated genes and proteins. In vivo, AAP/NAC inhibited tumor growth in EF43.fgf4 and 4T1 triple-negative breast tumors. Flow cytometry of tumor-associated macrophages revealed that AAP/NAC selectively inhibited M2 polarization. The anti-tumor activity of high-dose AAP/NAC is lost in macrophage-depleted mouse syngeneic tumor models, suggesting a macrophage-dependent mechanism of action. In conclusion, our study is the first to show that high-dose AAP/NAC has profound effects on the tumor immune microenvironment that facilitates immune-mediated inhibition of tumor growth.

## 1. Introduction

High-dose acetaminophen (AAP) has been evaluated as an anti-cancer agent both pre-clinically [1,2] and in early phase clinical studies with N-acetylcysteine (NAC) rescue to prevent liver toxicity [3,4]. A high objective response (21%) was observed in patients refractory to multiple lines of standard treatment in a phase I clinical trial of high-dose AAP with NAC rescue without any dose-limiting toxicities at doses as high as 20 g/m^2^ [3]. The lack of further clinical evaluation is likely due in part to the impracticality of oral administration and a lack of mechanistic understanding of AAP’s anti-cancer effects. In multiple different models, we have unequivocally established that liver toxicity, but not anti-cancer effects, is mediated through glutathione depletion secondary to the selective expression of CYP2E1 in zone 3 of the liver [5,6]. However, if the anti-tumor activity of high-dose AAP is not mediated by glutathione depletion, then an alternate mechanism must exist. To this end, we recently demonstrated that high-dose AAP acts as a novel inhibitor of STAT3 via direct binding [5]. More importantly, AAP has a high degree of binding specificity to STAT3 relative to STAT1 [5].

JAK/STAT signaling is known to play a central role in many elements of the immune system, and in particular, helps to mediate an effective anti-tumor immune response [7]. IL-4/STAT6 signaling in macrophages leads to an immune-suppressive M2 phenotype characterized by increased expression of arginase, an enzyme that metabolizes arginine into ornithine and urea. Ornithine plays a role in many repair processes including cell proliferation and collagen synthesis [8]. Tumor associated macrophages are considered to have a pro-tumorigenic M2 phenotype [9]. 

While M2 macrophages promote cellular repair, M1 macrophages promote an inflammatory response with anti-tumor cytotoxic effects [8]. M1 macrophages are characterized by the metabolism of arginine to nitric oxide via iNOS (inducible nitric oxide synthase), contributing to a tumor-suppressive pro-inflammatory microenvironment [10]. iNOS is strongly induced by interferon gamma and LPS via STAT1-mediated signaling [11]. 

In the present manuscript, we unequivocally show that high-dose AAP with NAC rescue may support favorable M1 differentiation while inhibiting the immune-suppressive M2 polarization of tumor-associated macrophages. Also, for the first time, we demonstrate that AAP is a selective inhibitor of pro-tumorigenic STAT6 while sparing anti-cancer STAT1 signaling.

## 2. Methods

**Bone marrow macrophage differentiation**. C57bl/6 mice were sacrificed, and their femurs and tibias placed in bone marrow medium (dulbecco’s modified eagle medium, DMEM, with 10% fetal bovine serum, FBS, and 25 ng/mL M-CSF from peprotech). Both ends of the femurs and tibias were cut, and a 27-gauge needle containing phosphate-buffered saline (PBS) was injected into the bone and collected on opposite side of bone containing the bone marrow cells. The bone marrow was then passed through a 70 µM filter and counted. The bone marrow cells were plated at 10^6^ cells per well of a 6-well plate in bone marrow medium. Every 3 days, the medium was replaced with fresh bone marrow medium. On about day 5, the bone marrow cells were differentiated into M2 macrophages using 20 ng/mL IL-4 or M1 macrophages using 20 ng/mL interferon gamma and 100 ng/mL LPS. Treatment with cytokine was for 24 h unless otherwise indicated. 

**Immunoblotting**: Bone marrow-derived macrophages (BMDM) were treated with NAC and/or AAP along with indicated cytokine(s). Cells were collected using RIPA buffer containing protease and phosphatase inhibitors and analyzed using Western blotting, using our previously published protocols [5].

**Quantitative PCR**: BMDM were left unpolarized (M0) or polarized to M1 or M2 phenotypes using appropriate cytokine(s). For the final 24 h prior to collection, EF43.fgf4 cells were grown on the same plate, separated from the macrophages using a transwell system with 0.4 μm sized pores (Corning, Corning, NY, USA), thus permitting chemical interactions between the tumor cells and macrophages, and thus mimicking the tumor microenvironment. The RNA was isolated and collected using TRizol reagent (Thermo Fisher, Waltham, MA, USA) per the manufacturer protocol. cDNA was created and quantified using our previously published protocols [5]. Primer pairs are listed in Appendix A.

**Cell culture**. LLC-luc cell lines were obtained directly from ATCC. EF43.fgf4 cells were obtained from the lab of Drs. Renata Pasqualini and Wadih Arap [12]. The 4T1 cells were obtained from collaborator Joseph Landry. Cell lines were used at low passage (<10) and periodically validated using morphology. All cell lines were grown in DMEM media with 10% FBS and antibiotic/antimycotic. Cell lines were periodically checked for mycoplasma and confirmed as mycoplasma free.

**Phagocytosis experiments**. BMDMs were cultured as described above. Tumor cells (EF43.fgf4 or LLC-Luc) were stained with Cellbrite Green (Biotium, Fremont, CA, USA) per manufacturer’s protocol. Stained tumor cells were added to macrophages for indicated amount of time. Tumor cells and macrophages were collected and stained for CD45 (AF700 BioLegend, San Diego, CA, USA, cat 103128), F4/80 (EF 450 Invitrogen cat 48-4801-80) and/or CD11B (BioLegend PERCP CY5.5, cat 101226) (as indicated in the Figure 4) in order to differentiate tumor cells from macrophages. Macrophages staining positive for both macrophage markers and cellbrite green (FITC channel) were considered to be phagocytic [13]. 

**Flow cytometry**. Flow cytometry experiments were performed on a BD Canto flow cytometer. Measurement of CD206 was assessed using a Cytek Aurora flow cytometer at the VCU Flow Cytometry Core Facility. BMDMs were cultured together with tumor cells as indicated. After treatment, cells were trypsinized, collected and stained using standard protocols. Tumor cells were distinguished from macrophages using CD45, CD11B and/or F4/80 stains and gating was performed accordingly. Live/dead gating was performed using Zombie Aqua (Biolegend, San Diego, CA, USA). Intracellular staining was performed using a kit from ThermoFisher, Waltham, MA, USA (Catalogue 00-5523-00). In vivo mouse tumor samples were homogenized using scissors and scalpels and then placed in a solution of Liberase TL Roche. Digestion media also contained 100 U/mL DNaseI (Sigma-Aldrich, St. Louis, MO, USA). Samples were mechanically digested in GentleMACS Dissociator and then incubated for 30 min at 37 deg F in digestion media. Cells were passed through a 70 µM cell strainer and RBC lysis was performed. Cells were stained for analysis using the following antibodies: MHC I APC (Invitrogen 17-5958-80), INOS AF488 (Invitrogen 53-5920-80), CD80 PE-CY7 (Invitrogen 25-0801-80), CD206 EF450 (Invitrogen 48-2061-80), PD-L2 PERCPCY5.5 (BioLegend 107217), Arginase PE (Invitrogen 12-3697-82), PD-L1 PE-CY7 (BioLegend 155405) and Arginase CD64 (Invitrogen 12-0641-82).

**Microscopy**. Fluorescent microscopy was performed in the VCU Microscopy Core Facility. BMDMs and tumor cells were cultured in a Lab-Tek II chamber slide (Fisher 154534). Cells were fixed and permeabilized using an intracellular staining kit from Thermo-Fisher (00-5523-00). Conjugated antibodies, listed in “Flow Cytometry” above, were used to stain the slides. Cells were imaged with an inverted Zeiss LSM880 confocal microscope, built on Axio Observer Z1 inverted stand and a 40× Plan Apo oil immersion objective (NA 1.4). Fluorescence was excited with 405 nm diode (15 Mw), 488 nm argon ion (25 Mw) or 633 nm He-Ne (3 Mw) laser. Laser power was adjusted (0.3–6%) so that the dynamic range was filled in 50% with the detector (PMT) gain of 550–650 (depending on the fluorochrome). Emission was detected in the following bands: 410–475 nm (DAPI), 490–550 nm (Alexa 488) and 645–740 nm (Alexa 647). The pinhole was set to 1.0 Airy units at 550 nm emission. Series of optical sections (z-stacks) were registered with 0.42 μs pixel dwell time (4× line averaging) and 16-bit precision. Pixel size was set to 71 nm and the z-step to 350 nm, respectively. Unless otherwise indicated, the optical magnification was 40× (oil objective), NA. The stacks of optical sections (DAPI, Alexa 488 and Alexa 647) were subjected to iterative deconvolution (maximum likelihood estimation) deconvolution (15–25 iterations) using Huyghens v. 22.04 (SVI, Hilversum, The Netherlands). The processing was initialized with the nominal PSF of the objective and performed at SNR set to 8–12, depending on the channel. The background was estimated as a minimum of average intensity in 15 × 15 pixel region.

**Mouse studies**. Tumor cells (EF43.fgf4 and 4T1) were cultured in standard media and injected orthotopically into the mammary fat pad of female BALB/c mice aged 8–10 weeks old. When tumors became palpable, mice were treated 2×/week with acetaminophen (Sigma, St. Louis, MO, USA) dissolved in 10% propylene glycol (PG) (Sigma) and 10% glucose (Sigma) in sterile water. N-acetylcysteine (Sigma) was dissolved in sterile water with 10% glucose. Macrophage depletion was performed using clone CI:A3-1 from BioXCell, and isotype was clone LTF-2 [14]. Antibodies were administered 200 µg weekly for 3 doses IP. Tumor size was monitored with digital calipers using the equation ½ (Length × Width^2^). All animal procedures were approved by the IACUC committee at the Richmond VA Medical Center.

**ELISA**. Tumors were collected in RIPA buffer containing protease inhibitors. Protein levels were normalized using a BCA Assay (Pierce Biosciences, Waltham, MA, USA). ELISA was performed using kits from BioLegend (Cat 430801 and 431411) according to the manufacturers protocol. Serum acetaminophen levels were assessed using an ELISA kit obtained from Immunalysis, following manufacturer’s protocol.

**Statistical analysis**. All data are presented as mean ± S.D. The Student’s *t* test was used for the comparison of measurable variants between 2 groups. Two-way analysis of variance (ANOVA), followed by the Tukey test, was used to evaluate differences among more than 3 groups. Two-way ANOVA, with Bonferroni post-test, was used for statistical comparisons between groups in tumor growth. *p* ≤ 0.05 was considered statistically significant (GraphPad Prism 6.0; Graph Pad Software).

## 3. Results

**AAP/NAC is not toxic towards normal organs:** In vivo, the treatment with high-dose AAP (500 mg/kg) given simultaneously with NAC/PG was well tolerated by mice. Of note, PG, in addition to being a diluent to enhance AAP solubility for in vivo administration, also helps reverse AAP toxicity via CYP2E1 inhibition [15] and thus was included in the rescue cocktail. AAP treatment alone led to prototypical zone 3 necrosis (around the central veins), where CYP2E1 enzymes are most concentrated [16]. A NAC-based rescue cocktail effectively rescued AAP hepatotoxicity (Figure 1A). Additionally, histologic evaluation of the lung and myocardium suggested no evidence of toxicity of AAP treatment in combination with NAC rescue (Figure 1A). 

To further evaluate the safety of AAP/NAC, mice were treated twice weekly for 2 weeks with NAC alone (100 mg/kg) or AAP (500 mg/kg) along with concurrent NAC. At 24 h after thefinal treatment, mice were sacrificed, and ALT and BUN levels analyzed. Results demonstrated that mice treated with AAP/NAC had no evidence of liver or kidney toxicity (Figure 1B) as evidenced by no change in the ALT or BUN levels compared to the NAC-alone treated mice.

**AAP selectively inhibits M2 vs. M1 polarization**: We next sought to study the effects of AAP/NAC on JAK/STAT signaling in macrophages. BMDMs stimulated with IFN gamma and LPS had no inhibition of STAT1 phosphorylation with the AAP/NAC treatment. On the other hand, BMDM stimulated with IL-4 had a substantial inhibition of STAT6 phosphorylation with the AAP treatment and this effect was not reversed by concurrent treatment with NAC (Figure 2A). 

In order to evaluate whether AAP/NAC may functionally inhibit BMDM polarization, effects of AAP/NAC on expression of a variety of M1 versus M2 genes (Figure 2) and proteins (Figure 3) were evaluated. RNA levels of IL-6, TNF alpha and IL-12 were all selectively induced by IFN gamma/LPS but not IL-4, suggesting that these genes are selectively expressed in M1-polarized macrophages. AAP/NAC did not inhibit expression of IL-12, IL-6 and TNF alpha (Figure 2B), and in fact, increased expression of IL-6 and TNF-alpha. On the other hand, known M2-associated genes CCL-24, YM1, arginase, FIZZ1, IRF4 and PD-L2 were all selectively induced in IL-4-stimulated M2 macrophages (and not induced in LPS/IFN gamma-stimulated M1 macrophages). Their expression was inhibited by AAP/NAC treatment (Figure 2C). 

We next evaluated the effect of AAP/NAC on M1 and M2-associated protein expression. Tumor cells (EF43.fgf4 or LLC-Luc, as indicated in Figure 3) were co-cultured with BMDM that were either undifferentiated (M0) or differentiated to either the M1 or M2 phenotype. It was demonstrated that M2-associated protein expression on F4/80+ macrophages (PD-L2, CD206 and arginase) was inhibited by AAP (Figure 3A–D). Additionally, fluorescent microscopy similarly revealed positive arginase staining in a subset of IL-4-treated BMDM and the number of arginase-positive macrophages was reduced with AAP/NAC treatment (Figure 3C). On the other hand, expression of M1-associated proteins INOS, MHC I, CD64 and PD-L1 was strongly induced by LPS/IFN gamma and was largely preserved with AAP/NAC treatment (Figure 3E,F). Of note, expression of IL-4 inducible proteins arginase and PD-L2 was not significantly changed by IFN gamma/LPS treatment. Similarly, M1-associated proteins PD-L1, MHC I, and INOS were minimally induced by IL-4 (Appendix A). Fluorescent microscopy demonstrated similar expression levels of the M1 marker CD64 [17] in vehicle and AAP/NAC-treated M1 macrophages (Figure 3F).

**AAP increases anti-tumor phagocytic activity:** We next sought to evaluate whether AAP/NAC may phenotypically affect BMDM using a phagocytosis assay [13]. Tumor cells (EF43.fgf4) were stained with Cellbrite Green and co-cultured with BMDM in the indicated treatment. F4/80 and Cellbrite Green co-positive cells were considered to be phagocytic macrophages (e.g., macrophages that had consumed a tumor cell). There was a marked increase in phagocytic macrophages with the AAP/NAC treatment (Figure 4A,B). Fluorescent microscopy demonstrated that macrophages treated with AAP/NAC had relatively large amounts of internalized fluorescent green stain (Figure 4C). On the other hand, BMDM treated with NAC alone had decreased phagocytosis of Cellbrite green labeled tumor cells. Given the possibility that the phagocytosis was simply from cytotoxicity of AAP towards tumor cells—indeed AAP/NAC was profoundly more toxic towards tumor cells relative to macrophages (Appendix A)—the experiment was repeated in the absence of treatment. BMDM were unpolarized (M0) or polarized towards the M1 or M2 phenotype overnight in conjunction with vehicle, NAC, or AAP/NAC. The media was washed off and replaced with serum-free media that contained no treatment or cytokine. Cellbrite Green stained tumor cells were then added and phagocytosis measured 4 h later. Increased phagocytosis was observed with AAP pre-treated macrophages for the M0, M1 and M2 conditions (Figure 4D).
Figure 4AAP increases macrophage phagocytosis. (**A**,**B**) Bone marrow-derived macrophages (BMDM) were cultured overnight with EF43.fgf4 breast cancer cells stained with Cellbrite Green along with indicated treatment (Veh, AAP 1 mM and/or NAC 0.3 mg/mL). Cells were analyzed by flow; F4/80 positive cells that were also Cellbrite Green-positive were considered phagocytic macrophages and were quantified as a percentage of total cells. Average of 3 independent replicates are shown in B. FMO (full minus one) control does not contain F4/80 antibody, only Cellbrite Green-positive tumor cells. (**C**) Representative images obtained using fluorescent microscopy. (**D**) BMDMs were cultured overnight with indicated treatment (Veh, AAP and/or NAC) and polarized to M0 (no cytokines), M1 (20 ng/mL IFN gamma and 100 ng/mL LPS) or M2 (20 ng/mL IL−4) macrophages overnight. The following day, all treatments were removed, and BMDM were placed in serum-free media. Cellbrite Green stained LLC-luc tumor cells were added for 4 h, and then fraction of phagocytic macrophages were analyzed using flow cytometry. * ≤ 0.05. Error bars represent ±1 SD.
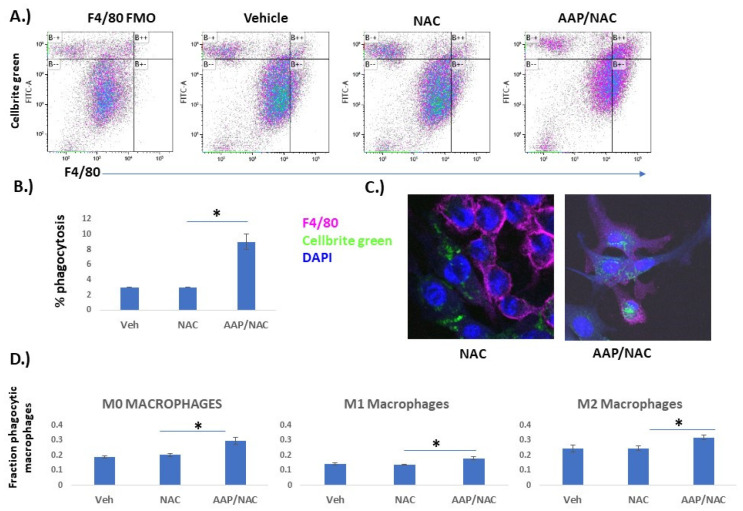



**AAP inhibits tumor growth and induces favorable macrophage polarization in vivo.** We next sought to evaluate the effects of AAP/NAC treatment on tumor-associated macrophages in vivo. EF43.fgf4 breast cancer tumors are known to have an immune infiltrate rich in macrophages [18]. The AAP/NAC treatment resulted in >50% reduction in tumor growth (Figure 5A,B). The protein lysate from the harvested AAP/NAC-treated tumors had a profound reduction in the M2-associated cytokine IL-10. The M1-associated cytokine interferon gamma was also decreased but to a lesser extent in the AAP/NAC-treated tumors (Figure 5C). Flow cytometry of the tumor immune microenvironment revealed a decreased expression of the M2 marker arginase in tumor-associated macrophages but an unchanged expression of M1 markers INOS, CD80 and MHC I (Figure 5D,E, Appendix A) and an increase in M1 marker CD64 (Appendix A). 

The effect of AAP/NAC on tumor associated macrophages in the triple-negative breast cancer tumor model 4T1 was also studied. AAP/NAC resulted in decreased tumor growth of 4T1 tumors (Figure 6A,B). Cytokine ELISA showed decreased levels of IL-10 but stable levels of IFN gamma in the tumor microenvironment (Figure 6C). Flow cytometry revealed decreased expression of arginase and CD206 and stable levels of INOS and MHCI in tumor-associated macrophages (Figure 6D,E, Appendix A). In both EF43.fgf4 and 4T1 models, there was at least a strong trend toward an increased macrophage content in the tumors (Appendix A). 

In order to evaluate the role of macrophages in mediating the tumor growth inhibition of AAP/NAC, 4T1 tumor-bearing mice were given macrophage-depleting antibodies or isotype control and the anti-cancer activity of AAP/NAC was evaluated. The anti-tumor effect of AAP/NAC was lost in macrophage-depleted mice (Figure 7), thus emphasizing the central role of AAP modulation of macrophage polarization in mediating the activity of high-dose AAP/NAC in our tumor models.

## 4. Discussion

High-dose AAP with NAC rescue demonstrated evidence of activity (3/14 assessable patients with response [3]) in a phase I trial without reaching a dose-limiting toxicity. Drugs with a 20% response rate or higher in phase I (less than 10% of drugs evaluated in phase I) have a 51% chance of meeting primary endpoints in subsequent phase II trials [19]. The aim of the present research is to gain mechanistic insights that may facilitate future clinical evaluation of high-dose AAP with NAC rescue as a novel anti-cancer therapy.

One of the major benefits of immunotherapies over traditional cytotoxic chemotherapy is the duration of benefit. Upon treatment with chemotherapy, tumor cells almost inevitably mutate leading to the emergence of resistant clones, a process contributing to systemic relapse. On the other hand, in the setting of an anti-tumor immune response, when the tumor mutates, the immune system can effectively adapt and evolve. The end result is often sustained responses and sometimes even cures [20]. While AAP is not thought to have anti-inflammatory properties at traditional FDA-approved doses, the present manuscript is the first to provide evidence that high-dose AAP with NAC rescue modulates the anti-tumor immune response.

We demonstrate that AAP/NAC inhibits IL-4/STAT6 preferentially over LPS/IFN gamma/STAT1 signaling, resulting in inhibition of the M2 polarization of macrophages. AAP/NAC inhibits the expression of several different M2 markers to a variable extent, likely a result of the complex interplay of competing signaling pathways. While JAK/STAT signaling plays a central role in macrophage polarization [21], multiple other pathways have been implicated as well, including NFκB signaling [22]. One unanswered question from the present study is the mechanism by which AAP/NAC modulates JAK/STAT signaling in macrophages. We previously demonstrated that AAP directly binds to STAT3 with a high degree of specificity relative to STAT1 [5]. Whether AAP inhibits IL-4/STAT6 signaling via direct binding to STAT6 versus indirectly via modulation of upstream signaling pathways is an active area of investigation in our laboratory. 

The tumor immune microenvironment is derived from the complex interplay of multiple distinct elements from the innate and adaptive immune systems. Importantly, JAK/STAT signaling plays a central role in the differentiation of multiple facets of the immune system, most notably T-cells [23] and dendritic cells [24]. It is unlikely that AAP mediates its anti-tumor activity exclusively via macrophage polarization; our lab is working to comprehensively characterize the effects of AAP/NAC on the interplay of the various components of the immune system involved in an effective anti-tumor immune response.

Our identification of macrophages as key intermediaries of the anti-tumor activity of AAP/NAC may have important implications for the selection of patients in future clinical trials. For instance, patients with macrophage rich tumors—a characteristic that could potentially be identified non-invasively using ferumoxytol-contrasted MRI [13]—may derive increased benefits from an AAP/NAC-based treatment.

## 5. Conclusions

In conclusion, in the present work, we identify a novel free-radical independent mechanism of anti-tumor activity of high-dose AAP with NAC rescue. We demonstrate that AAP/NAC inhibits IL-4/STAT6 signaling, and the resulting M2 polarization, in tumor-associated macrophages. This observation is the first to mechanistically link the anti-tumor efficacy of AAP/NAC with the anti-tumor immune response.

## Figures and Tables

**Figure 1 cancers-15-04770-f001:**
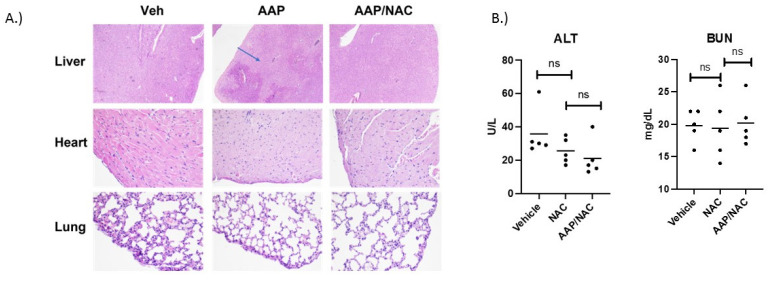
AAP/NAC is not toxic towards normal organs. (**A**) C57bl/6 mice were treated with a single dose of Vehicle, AAP alone (500 mg/kg) and AAP/NAC (100 mg/kg)/propylene glycol (PG) (10%) i.p. Mice were sacrificed the following day, and sections of liver (H&E 40×) and sections of heart and lung (H&E 200×) were examined. Blue arrow points to area of hepatocyte necrosis. (**B**) Mice were treated 2×/week for 2 weeks with NAC (in 10% PG) or AAP/NAC (10% PG) and 24 h after last treatment, serum BUN and ALT were measured.

**Figure 2 cancers-15-04770-f002:**
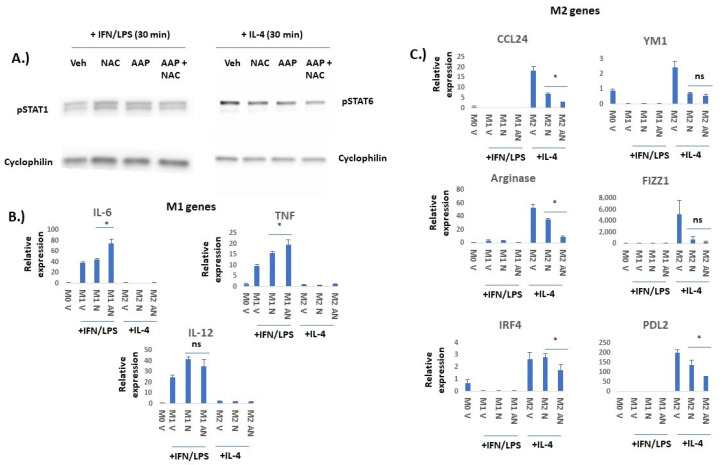
AAP selectively inhibits IL-4/STAT6-mediated M2 polarization and not IFN/LPS/STAT1-mediated M1 polarization in bone marrow derived macrophages (BMDM). (**A**) BMDM were treated for 6 h with Veh, AAP (1 mM) and/or NAC (0.3 mg/mL) and then stimulated with IFN gamma (20 ng/mL)/LPS (100 ng/mL) for 30 min prior to collection of protein lysate for immuno-blotting analysis. Similarly, BMDM were treated for 18 h with Veh, AAP (1 mM) and/or NAC (0.3 mg/mL) and then stimulated with IL-4 (20 ng/mL) for 30 min prior to collection of protein lysate for immuno-blotting analysis. (**B**,**C**) BMDM were treated with Veh (M0 macrophage), IL-4 (M2 macrophages) or IFN gamma/LPS (M1 macrophages) overnight along with vehicle (V), AAP (A) and/or NAC (N), using same concentrations as above. EF43.fgf4 cells were cultured with the BMDM for 24 h using a trans-well insert to physically separate the tumor cells from the BMDM. BMDM were collected and analyzed using qPCR. Values normalized to M0 (unpolarized) vehicle-treated macrophages. Error bars represent ± 1 SD * ≤ 0.05, ns = not significant. Original western blots are presented in Appendix A.

**Figure 3 cancers-15-04770-f003:**
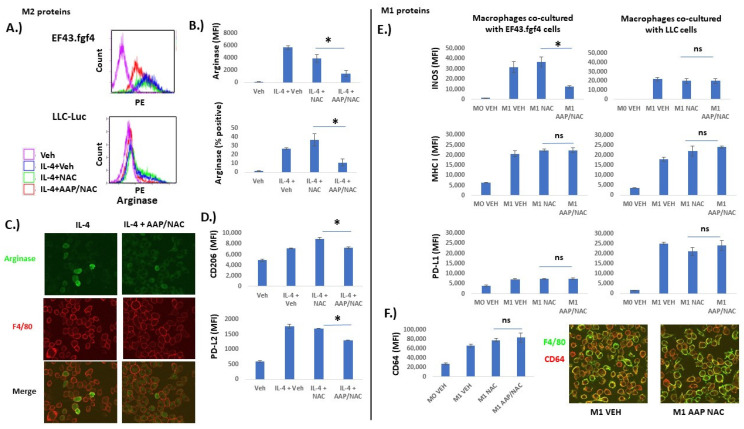
AAP selectively inhibits M2-associated protein expression relative to M1-associated protein expression. Bone marrow derived macrophages (BMDM) were left unstimulated (Veh), or stimulated overnight with IL-4 (20 ng/mL) to stimulate differentiation into M2 macrophages (**A**–**D**) or IFN gamma (20 ng/mL) and LPS (100 ng/mL) to stimulate differentiation into M1 macrophages (**E**,**F**). (**A**) Representative flow plots of macrophages that were co-cultured with indicated tumor cell line. Gating was for F4/80+ macrophages. (**B**) Average of 3 independent trials. (**C**) Macrophages co-cultured with EF43.fgf4 tumor cells overnight along with indicated treatment. Confocal microscopy was used to obtain representative images. (**D**) BMDM were co-cultured overnight with EF43.fgf4 cells along with indicated treatment and analyzed with FACS. Shown is average of 3 independent trials. (**E**,**F**) BMDM were co-cultured with indicated cell line (**E**) or EF43.fgf4 cells for 24 h along with no stimulation (M0) or IFN gamma and LPS (M1) along with indicated treatment. Analyses for protein expression was performed for 3 independent samples via flow cytometry, gating for CD45+/F480+ macrophages. (**F**) Confocal microscopy was used to image M1 marker CD64 on F4/80 positive macrophages. * ≤ 0.05. Error bars represent ±1 SD.

**Figure 5 cancers-15-04770-f005:**
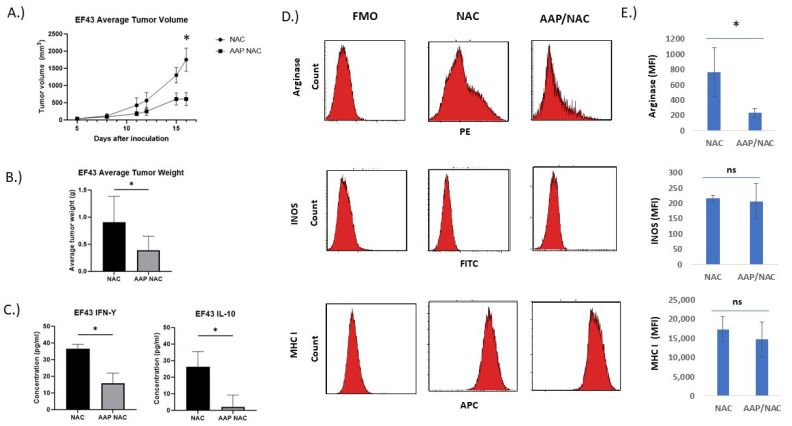
AAP inhibits EF43.fgf4 tumor growth in vivo. (**A**) BALB/c mice containing EF43.fgf4 (EF43) tumors were treated 2×/week with NAC (100 mg/kg)/Propylene glycol (PG, 10%) or AAP (500 mg/kg)/NAC/PG and tumor growth monitored with digital calipers. (**B**) At time of animal sacrifice, tumors were weighed. (**C**) At conclusion of study, tumors were digested and analyzed for IL-10 and IFN gamma by ELISA. (**D**) CD45+/CD11B+/F4/80+ macrophages were analyzed for M1 and M2 macrophage markers by flow cytometry. (**E**) Average MFI of indicated markers in macrophages from 3 mice tumors per treatment group are shown. * ≤0.05. Error bars represent ±1 SD.

**Figure 6 cancers-15-04770-f006:**
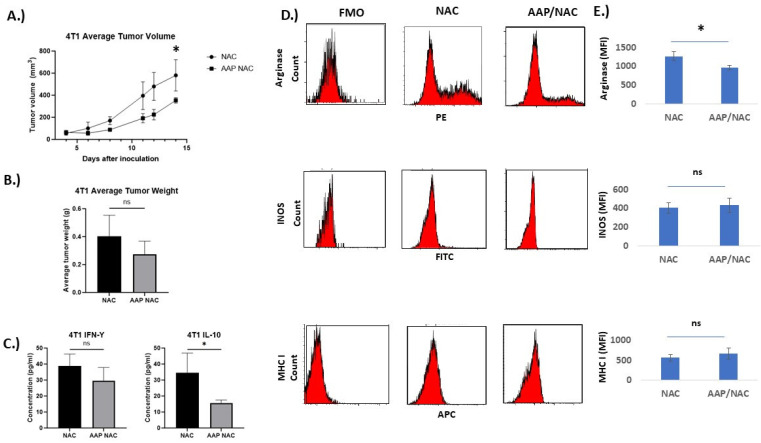
AAP inhibits 4T1 tumor growth in vivo. (**A**) BALB/c mice containing 4T1 tumors were treated 2×/week with NAC (100 mg/kg)/propylene glycol (PG, 10%) or AAP (500 mg/kg)/NAC/PG and tumor growth was monitored with digital calipers. (**B**) At time of animal sacrifice, tumors were weighed. (**C**) At conclusion of study, tumors were digested and protein levels of IFN gamma and IL-10 determined using ELISA. (**D**) CD45+/CD11B+/F4/80+ macrophages were analyzed for M1 and M2 macrophage markers by flow cytometry. (**E**) Average MFI of indicated markers in macrophages from 3 mice tumors per treatment group are shown. N = 3 per condition. * ≤0.05. Error bars represent ±1 SD.

**Figure 7 cancers-15-04770-f007:**
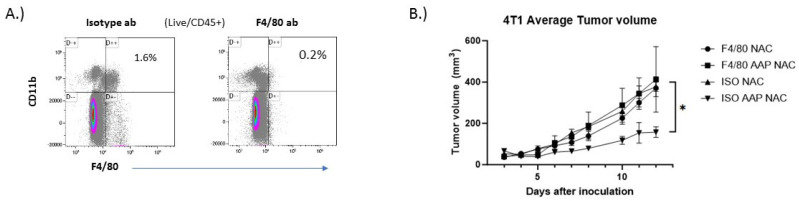
Anti−cancer activity of AAP/NAC reversed by macrophage depletion. (**A**) BALB/c mice were treated with 200 µg of F4/80 depleting antibody or isotype control and 6 days later, sacrificed and spleen analyzed for macrophage content. (**B**) BALB/c mice bearing 4T1 breast tumors were given macrophage-depleting antibody (F4/80) or isotype control (ISO) along with NAC (100 mg/kg)/propylene glycol (PG, 10%) or AAP (500 mg/kg)/NAC/PG and tumor size monitored with digital calipers. N = 5 per condition. * ≤0.05, ISO AAP NAC is different from other three groups by two-way ANOVA. Error bars represent ±1 SD.

## Data Availability

All data available upon request.

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
