# Peer review of "High-Dose Acetaminophen with N-acetylcysteine Rescue Inhibits M2 Polarization of Tumor-Associated Macrophages"

_cancers, 2023, doi:10.3390/cancers15194770_

Round 1
Reviewer 1 Report
This is an interesting study where the author found that AAP/NAC has the ability to regulate macrophage polarilization and thus have anti-tumor effects. Although it is a complete story, it can still be more fulfilling. The author also mentioned that the relationship between AAP and the JAK/STAT signaling pathway has not been included, and I believe this part could be supplemented to make the story more complete.
In addition, I have a few small questions: 1, why choose triple negative breast cancer cells to study the role of AAP/NAC. 2. What is the impact of excessive NAC? Is there data for NAC treating group?
Author Response
Critique: The author also mentioned that the relationship between AAP and the JAK/STAT signaling pathway has not been included, and I believe this part could be supplemented to make the story more complete.
Response: We thank the Reviewer for the critique. We have previously shown that AAP binds STAT3 with higher affinity relative to STAT1 (Pingali et al, Neoplasia). In the present manuscript, we demonstrate that AAP inhibits STAT6 but not STAT1 (Figure 2), however we have not yet elucidated the mechanism. Effective validation of the mechanism would require extensive additional experimentation (in vitro binding studies, in vitro knockdown studies, potentially in vivo KO studies) that we believe is outside the scope of the current manuscript.
Critique: Why choose triple negative breast cancer (TNBC) cells to study the role of AAP/NAC.
Response: As we have recently published, EF43.fgf4 triple negative breast cancer cells form highly inflammatory tumors with a large macrophage infiltration (Sillerud et al, Cancers, 2021). As a result, we felt TNBC would be an ideal model for our macrophage studies.
Critique: What is the impact of excessive NAC? Is there data for NAC treating group?
Response: In our studies, we use 100 mg/kg NAC IP. This is because starting at 400 mg/kg NAC IP, there is toxicity in mice--ALT and BUN are markedly elevated when measured the following day. Intermediate doses, eg 200 mg/kg NAC, don't add much benefit in our experience over 100 mg/kg.
NAC-based rescue cocktails alone have no anti-tumor activity relative to an untreated vehicle (data not shown). Because the goal of our study is to evaluate the effects of AAP, we use NAC-based cocktails as our vehicle control group.
Reviewer 2 Report
The manuscript is acceptable, however, some problems require careful consideration.
1. Literature need to be updated.
2. The figure resolution needs to be improved.
3. In supplementary materials, a presentation of parallel experimental results should be provided.
Author Response
Critique: Literature need to be updated.
Response: We thank the Reviewer for this recommendation. We have reviewed the references and made updates as indicated, including adapting the formatting to MDPI style.
Critique: The figure resolution needs to be improved.
Response: We have replaced all of the figures with JPEG files in hopes that this will improve the resolution of the images.
Critique: In supplementary materials, a presentation of parallel experimental results should be provided.
Response: Our supplementary materials, attached, include parallel experiments to support our conclusions.
We thank the Reviewer for the critiques.
Reviewer 3 Report
The article by Bryan et al. concerns investigation of mechanism of action paracetamol/N-acetylcysteine mixture as antiproliferative agent. The article is well written, it is in high degree easy to understand the material; there are many Figures (which are useful for material perception). I would like to emphasize the simple system used for possible application in practice; the undesired effects of this agents are significantly weak (especially in comparison with widely used nowadays). I support the Acceptance of this Manuscript after Minor Revision (see Notes below).
Notes for the Authors:
1) Change title to “N-acetylcysteine” (acetylated amino-group).
2) Line 35: syngeneic mice?
3) Line 44: doses in g/m2? It is very strange size dimension.
4) Line 73: decipher abbreviations the first time they are mentioned. Check all text.
5) Check the reference list for the rules of the journal and abbreviations (https://cassi.cas.org/search.jsp).
Author Response
Critique: Change title to “N-acetylcysteine” (acetylated amino-group).
Response: This change has been made. Thank you.
Critique: Line 35: syngeneic mice?
Response: We have revised this sentence to say "mouse syngeneic tumor models." Thank you.
Critique: Line 44: doses in g/m2? It is very strange size dimension.
Response: Dr. Kobrinski's study did indeed use g/m2 to report his AAP dosing. m2 represents body surface area, a commonly used measure of patient size to dose chemotherapy-type agents.
Critique: Line 73: decipher abbreviations the first time they are mentioned. Check all text.
Response: We have reviewed all text and defined abbreviations first time they are used.
Critique: Check the reference list for the rules of the journal and abbreviations (https://cassi.cas.org/search.jsp).
Response: We have reviewed the reference list to make sure all references follow rules of the Journal.
Thank you for the helpful suggestions.